# Strong Bioactive Glass-Based Hybrid Implants with Good Biomineralization Activity Used to Reduce Formation Duration and Improve Biomechanics of Bone Regeneration

**DOI:** 10.3390/polym15173497

**Published:** 2023-08-22

**Authors:** Jing Chen, Yonglei Xing, Xiaozhuan Bai, Min Xue, Qi Shi, Beibei Li

**Affiliations:** 1The Key Laboratory for Surface Engineering and Remanufacturing in Shaanxi Province, School of Chemical Engineering, Xi’an University, Xi’an 710065, China; xiaozhuanBai@163.com (X.B.); xuemin@xawl.edu.cn (M.X.); shiqi@xawl.edu.cn (Q.S.); beibeili0813@xawl.edu.cn (B.L.); 2State Key Laboratory of High-Efficiency Utilization of Coal and Green Chemical Engineering, National Demonstration Center for Experimental Chemistry Education, School of Chemistry and Chemical Engineering, Ningxia University, Yinchuan 750021, China

**Keywords:** bioceramics, modified cellulose nanofibers, bioactive glass, biomineralization activity

## Abstract

Developing bioactive implants with strong mechanical properties and biomineralization activity is critical in bone repair. In this work, modified cellulose nanofiber (mCNF)-reinforced bioactive glass (BG)-polycaprolactone (PCL) hybrids (mCNF–BP) with strong biomechanics and good apatite formation ability were reported. Incorporating mCNFs shortens the forming duration of the hybrid films and enhances the biomechanical performance and in vitro apatite-formation capability. The optimized biomechanical performance of the optimal hybrid materials is produced at a relatively high mCNF content (1.0 wt%), including a considerably higher modulus of elasticity (948.65 ± 74.06 MPa). In addition, the biomineralization activity of mCNF–BP hybrids is also tailored with the increase in the mCNF contents. The mCNF–BP with 1.5 wt% and 2.0 wt% mCNFs demonstrate the best biomineralization activity after immersing in simulated body fluid for 3 days. This study suggests that mCNFs are efficient bioactive additive to reinforce BG-based hybrids’ mechanical properties and biomineralization activity.

## 1. Introduction

The elderly suffering trauma, revision surgery, and comminuted fractures often lead to large-area non-self-healing bone defects. For repairing large-area bone defects, bone graft substitutes are required to provide mechanical support, enhanced bioremediation after bone injury, and efficient surface area for cell attachment, proliferation, and the resulting bone regeneration [1,2]. The most efficient methods of extensive bone repair include bone grafts, osteotomy by traction, etc. Traditional synthetic transplantation materials, such as bioceramics, biopolymers, and composites, have been adopted to substitute autografts and allografts, which are unavailable and at risk of immunogenicity [3]. There is an imminent need to innovatively design bone graft or bone-repairing materials, which should be biocompatible, osteoconductive, bioactive, processable, and exhibit strong mechanical properties. 

Compared to pure polymers and bioceramics, nanocomposite materials are promising to meet the various requirement of novel bone-repairing biomaterials [4]. For example, the organically modified sol-gel bioceramics, termed bioactive glass (BG), are highly flexible, mechanical-strength-controlled, and bioactivity-tailored [5,6]. BG can form a bone-like apatite-mineral phase when contacting living tissue. Its unique advantages include its elastomeric property, biodegradability and biocompatibility, which facilitate its broad application to bone-repairing [7]. Bioactive glass nanoparticles (BGN) are introduced to the polymer matrix to prepare bioactive nanocomposites, which have been proven as a promising method [8]. The advantages of polymers, such as elastomery, biodegradability, biocompatibility, and easy processing ability, are also completely utilized [9,10]. However, the poor compatibility between inorganics and polymers leads to unreliable mechanical properties and the unexpected heterogeneous biomineralization activity of the BGN-based nanocomposites [11,12]. To solve this problem, nanofibers have been employed to enhance the cell attachment and increase the volume ratio by sufficient pore interconnectivity [13], which provides a biocompatible interface ability and reinforcing ability by the inherent strength [14,15]. Cellulose nanofibers (CNFs), promising nanofibers, have attracted much attention due to their specific surface area, high aspect ratio, high stiffness, and high mechanical strength. With these prominent advantages, CNFs as a reinforcement filler are widely incorporated in polymeric hydrogel matrices [16,17]. The hybrid materials synthesized with inorganic nanoparticles and a small number of CNFs can result in synergistic effects and induce high ductility and toughness with novel deformation mechanisms [18]. However, CNFs have some significant disadvantages, such as poor interfacial adhesion. The pristine CNFs physically incorporated with the matrix will lead to weak bonding strength despite the resulting enhanced mechanical properties, which somewhat restricts the wide commercial application of CNF-reinforced composites [19,20]. CNF surface modification is another approach to creating covalent or noncovalent connections between nanofibers and the polymer matrix to produce stronger hydrogels [21]. In order to allow fillers to be well dispersed and adhered to the polymer matrix, surface modifiers, such as surfactants, titanate coupling agents, and silane coupling agents, are required. This increases the compatibility between the fillers and the organic matrix, as expected. Additionally, some functional groups, such as silyl, aldehyde and carboxylate, could be used to modify the surface of CNFs. Silane coupling agents can form strong chemical bonds with polymer materials to enhance the compatibility of composites significantly by modifying interfacial adhesion [22]. Chen et al. found that the silane coupling agent, KH560 (c-(2,3-epoxypropoxy) propyl trimethoxysilane), as a surface modifier was incorporated into the low-density polyethylene (LLDPE), which could effectively improve the properties of composites [23]. Above all, the strong bond between the filler and polymer matrix can improve interfacial adhesion by the functional modifier [24,25]. KH560 modulates cellulose nanofibers (mCNFs) by forming firm and stable chemical bonds to enhance the compatibility between CNFs and the polymer matrix. Our literature search indicates that no one has investigated these functionalized mCNFs’ impact on improving synthetic gels’ properties. Meanwhile, the content of mCNFs is manipulated and fine-tuned to strengthen their miscibility and interfacial compatibility with host matrices and to endow new functionalities to the hybrid materials.

In previous studies, we have successfully produced BG-Poly (ε-caprolactone) (PCL)-based (BP) monolith hybrids via a sol-gel method, which has a high apatite-forming ability. Adding PCL significantly improves the formation of nanofiber structures, and the hybrid membranes showed specific mechanical properties. Moreover, the sample of BP (30 wt%) exhibited the best mechanical properties, especially the highest elastic modulus [9,26]. To further improve the molding ability and mechanical properties of the material, in this work, we apply mCNFs as a reinforcing agent, due to the better mechanical properties of mCNFs. Therefore, mCNFs as a nano reinforcement are firmly entangled with the polymer matrix, improving hybrid membranes’ performance. A simple sol-gel approach is used to create crack-free mCNF–BP hybrid membranes. The effects of mCNF–BP hybrid membranes with varying mCNFs loading contents on biomineralization activity were investigated in this research. Incorporating mCNFs is predicted to significantly enhance the mechanical properties and biomineralization activity of mCNF–BP hybrid membranes.

## 2. Materials and Methods

### 2.1. Materials

Tetraethoxysilane (TEOS, Si(OC_2_H_5_)_4_), Calcium nitrite (Ca(NO_3_)_2_·4H_2_O), isopropyl-alcohol (IPA), tetrahydrofuran (THF), dichloromethane (DCM) and hydrochloric acid (HCl, 35%) were obtained from Guanghua Chemical Factory Co., Ltd. (Xi’an, China). KH560 modified cellulose nanofibers (mCNFs) were purchased from North Century cellulose material Co., Ltd. (Xuzhou, China). Polydimethylsiloxane (PDMS, HO-[Si (CH_3_)_2_-O-]nH, Mn = 1100) was supplied by Alfa (Alfa, Ward Hill, MA, USA). PCL ((C_6_H_10_O_2_) n) (Mn = 80,000) was purchased from Sigma-Aldrich (Sigma-Aldrich, St. Louis, MO, USA).

### 2.2. Synthesis of mCNF–BP Hybrids

The mCNF–BP hybrid bulk materials were synthesized by a modified sol-gel method. Briefly, TEOS (4.166 g) was dissolved in a mixed solvent of iso-propyl-alcohol (IPA) and tetrahydrofuran (THF) with a volume ratio of 1:1. After stirring for 30 min, the catalyst with 0.075 g of HCL and 0.36 g of water was added. Then, PDMS (1.788 g) was added to the solution for hydrolysis reaction for 2 h. After 20 h of reaction, the previous solution was mixed with Ca(NO_3_)_2_∙4H_2_O, IPA, and H_2_O to react for 1 h to produce the sol. The obtained sol BP (0.3 g) was added to 10 mL of DCM and stirred vigorously for 10 min. Then, the predetermined content of mCNFs (0.5, 1.0, 1.5, 2.0 and 2.5 wt% in relation to PCL) was sonicated in the former DCM intermixture for 30 min. Next, 1 g of PCL pellets were combined with additional violent stirring for 12 h. The prepared mixture was then transferred into Teflon dishes and kept at 30 °C for 1 day to obtain the mCNF–BP hybrid gel. Finally, the mCNF–BP hybrid monoliths were baked at 40 °C for 24 h.

### 2.3. Characterization

Surface morphologies of the mCNF–BP film were analyzed by scanning electron microscopy (SEM, JSM-6390, JEOL, Tokyo, Japan). The elemental composition was analyzed by energy dispersive spectroscopy (EDS, JEOL, Tokyo, Japan). The crystalline composition of the specimens was measured by the XRD (XRD, D/MAX-2400, Rigaku, Tokyo, Japan).

### 2.4. Biomineralization Activity Test

The apatite-forming ability of the samples was determined after soaking for 3 days in simulated body fluid (SBF) by observing the apatite generation on their films. The samples were cut into small pieces with a size of 10 × 10 × 2 mm^3^ and soaked in SBF (in mM: Ca^2+^ 2.5, Mg^2+^ 1.5, Na^+^ 142, K^+^ 5.0, Cl^−^ 147.8, SO_4_^2−^ 0.5, HCO_3_^−^ 4.2, HPO_4_^2−^ 1.0), which was similar to the case of human plasma. The samples were taken out from the solution after culturing in 30 mL SBF for 3 days at 36.5 °C. Then, the samples were cleaned with deionized water and dried at 37 °C for 1 day. Finally, SEM, EDS and XRD were applied to analyze the surface apatite-forming ability of samples.

### 2.5. Mechanical Property Test

The tensile properties of hybrid monoliths were tested by a universal mechanical machine with a 500 N load cell (SHT4206, MTS, Eden Prairie, MN, USA) with a crosshead speed of 50 mm min^−1^. The PCL and mCNF–BP were cut into small pieces with a size of 10 mm × 60 mm to perform the tensile mechanical test. The slope in the linear region of the stress–strain curve is used to determine the tensile module. Note that 5 samples per experimental condition were measured.

### 2.6. Hydrophilicity Measurement

Dynamic water contact angle measurement (SL200KB, Shanghai, China) was used to test the hydrophilicity of the samples. Cut the mCNF–BP into small pieces with a size of 10 mm × 40 mm to test the contact angles. Five points per specimen were tested. A 21-gauge needle was used to drop deionized water on the samples. The pictures were taken to measure the water contact angles.

### 2.7. In Vitro Cytotoxicity Tests

The osteoblastic biocompatibility of mCNF–BP was determined by evaluating the cellular in vitro toxicity of the MC3T3-E1 osteoblast cell line on the sample surface. In a humidified growth medium (MEM, Hyclone) at 37 °C and 5% CO_2_ with 10% fetal bovine serum (FBS, Gibico) and 1% penicillin/streptomycin (Hyclone) environment, Cell lines were cultivated. In 96-well plates, (MC3T3-E1) cells were seeded in 96-well plates at a density of 3000 well^−1^. After cell lines were cultured for 24 h, cell lines were exposed to mCNF–BP samples at concentrations of 50, 100, 150, and 200 µg mL^−1^ for 72 h. Cell lines on tissue culture plates in the same medium without BGN were used as a control. The media was withdrawn from each well at the preset time intervals (24 h and 72 h), then 10 µL of MTT (Sigma, St. Louis, MO, USA) reagent in PBS (5 mg mL^−1^) was added to each well, and wells were then incubated at 37 °C for 4 h. The formazan crystals were dissolved in DMSO and the optical density (OD) was then determined using a microreader (BIO-TEK, ELX800) at 490 nm. Untreated cells were used as controls, and the data were translated to cell viability with the following equation:

Relative cell viability (%) = ODs/ODc × 100%, where ODs and ODc represented the absorbance values of samples and control, respectively.

### 2.8. Statistics

Standard deviation (SD) was indicated for all statistics. The *t*-test analysis was used to measure the statistical differences. Statistical significance was displayed as * *p* less than 0.05 and ** *p* less than 0.01.

## 3. Results

Firstly, PDMS-BG is dissolved in DCM to form a sol solution. Regarding the mCNFs, KH560 is used to modify the surficial group to make the hydrophilic CNFs become hydrophobic. Then, by mixing the PDMS-BG solution and the mCNFs suspension in the PCL matrix uniformly, the mCNF–BP hybrids form as expected, as shown in Figure 1. Because of the filamentous structure of the mCNFs, the single fibers have uniform size and facilitate the processing of homogeneous suspension. The mCNFs, as a cross-linker and nano-reinforcing filler, exhibit a large specific surface area, which provides massive active junctions and thereby significantly enhances the gelation ability of the mCNF–BP hybrids.

The crystal structure of mCNF–BP hybrids with various mCNFs ratios (0.5, 1.0, 1.5, 2.0, 2.5 wt%) was evaluated by XRD. Figure 2 shows specific diffraction peaks at 21.88° and 23.85°, suggesting that PCL is a kind of semi-crystalline polymer. The weaker diffraction peaks 16.2°, 22.6°, and 34.6° correspond to the mCNFs. Because the BG is typically amorphous, there are no corresponding peaks observed.

The micro-morphology and structure of the mCNF–BP hybrids with various mCNFs ratios are shown in Figure 3. The pure BP exhibits a very smooth surface (Figure 3a). When mCNFs are incorporated, the resulting mCNF–BP hybrids exhibit a relatively rough surface (Figure 3b–f). Additionally, one can observe the homogeneously distributed filamentous structure on the PCL matrix surface. The more filamentous material is shown in higher added content of mCNF, representing the excellent combination between mCNFs and BP. Figure 4 shows the EDS spectra to reveal the chemical compositions of the mCNF–BP hybrids. The results confirm the presence of silicon (Si), Carbon (C), and Oxygen (O) in the matrix. As expected, the observed Si and Ca peaks suggest that the mCNFs-BG can be effectively hybridized into the PCL matrix.

### 3.1. Mechanical Performance Test

The tensile-strain behaviors of the PCL and mCNF–BP hybrids with various mCNFs incorporations (0, 0.5, 1.0, 1.5, 2.0, and 2.5 wt%) are illustrated in Figure 5a. When the mCNF incorporation is less than 2.0%, the samples showed uniform elastic deformation and brittle fracture. Figure 5b presents the ultimate tensile stress of the PCL and mCNF–BP hybrids. It is shown that the ultimate tensile stress of hybrid material increased from 19.37 ± 0.52 to 35.37 ± 3.29 MPa as the mCNF amount increased from 0 to 0.5 wt%. Then, the ultimate tensile stress decreases to 14.55 ± 1.85 MPa as the mCNFs amount increases from 0.5 to 2.5 wt%. The Young’s modulus of the PCL and mCNFs–BP hybrids, as shown in Figure 5c, exhibits a similar trend to the ultimate tensile stress. The Young’s modulus increased significantly from 467.95 ± 42.67 to 948.65 ± 74.06 MPa with the addition of different contents of mCNFs, and then decreased to 376.02 ± 32.91 MPa. In comparison, the maximum Young’s modulus of the PCL samples is 495.21 ± 52.33 MPa. The maximum tensile stress of BP with the addition of the exact content of PMS-BG is 19.37 ± 0.52 MP and the maximum Young’s modulus is 467.96 ± 42.67 MPa, the enhancement of the ultimate tensile stress and Young’s modulus when the mCNFs incorporation is less than 0.5 wt% is attributed to the restricted migration of the soft PCL chains, because mCNFs, as a reinforcement agent, entangle in the hydrogel network. In this regard, mCNFs may be a potential reinforcing material or cross-linker widely used to improve the mechanical properties of other biodegradable polymer materials. However, in the case of mCNF incorporation larger than 1.0 wt%, the reductions in the ultimate tensile stress and Young’s modulus are likely attributed to the agglomeration of mCNFs themselves, which cannot hybridize with the PDMS and PCL stably. Although the mCNFs incorporation leads to mechanical improvement to some extent, the strain at failure of the samples decreases as the mCNFs incorporation increases, due to the damage of the semi-crystalline structure of PCL, as shown in Figure 5d.

### 3.2. mCNF–BP Hydrophilicity Test

The excellent surface hydrophilicity of the biomaterial implants is crucial to the conformation of adsorbed proteins, which significantly impacts the biological response to bone tissue regeneration. Figure 6 shows the results of the water-contact-angle tests in the case of various amounts of mCNFs incorporation. The pure PCL exhibits a water contact angle of around 100° because of its hydrophobic property. In the case of the mCNF incorporation, the water contact angle decreases by around 20%, suggesting the increasing hydrophilicity compared to the pure PCL is likely attributed to the high hydrophilic group of the mCNF phase.

### 3.3. Biomineralization Activity Measurement

The biomineralization activity is crucial to bone conduction and bone regeneration abilities of the biomaterials. By soaking the mCNF–BP samples into SBF for 3 days, it is observed that the apatite forms in vitro, which is characterized by SEM, EDS, and XRD measurements. Previously, we found no noticeable apatite particles are generated on the pure PCL even after 7-day soaking in SBF. Regarding the mCNF–BP hybrids, the SEM results show the obvious layer-shaped apatite (Figure 7) compared to the non-soaking sample (Figure 3). The formation of *hydroxyapatite* is shown in Figure 7a–f, where calcium ions dissolved from the bioactive glass (BG) increase the supersaturation of the surrounding body fluids relative to the apatite. The hydrated silica formed on its surface provides a specific vantage point for apatite nucleation. Consequently, apatite nuclei readily form on CaO, SiO_2_-based glass surfaces and grow spontaneously by consuming calcium and phosphate ions from the surrounding body fluids. The minerals are deposited and aggregated on the surface of the sample in the form of a globular accumulation. As shown in Figure 7d,e, more apatite generates as the mCNFs incorporation increases because the increasing specific surface area and the aspect ratio of the mCNFs lead to the numerous active cross-linking junctions to enhance the biomineralization activity. Such a result suggests that the mCNF incorporation has a very significant impact on the apatite generation.

Figure 8 and Table 1 present the EDS spectra of the mCNF–BP hybrids with various amounts of mCNF incorporation after 3-day soaking in SBF. All the samples show the peaks of Ca and P. The peak areas as indexes of the element amount of Ca and P exhibit apparent increasing trends as the incorporation amount of mCNFs increases to 2.0 wt%, due to the improved apatite-formation ability induced by the extensive active junctions in the case of mCNFs in the hybrids. Based on the micrograph and EDS analysis, the 1.5 wt% of the mCNFs was uniformly coated with a dense layer of calcium phosphate and had a Ca/P weight ratio (1.95), which is much more similar to natural bone. However, the 2.5 wt% of the mCNF incorporation leads to agglomeration to reduce the dispersion ability of mCNFs and, thereby, the density of active junctions, which suppresses the apatite generation [19,24]. Consequently, the peak intensities of Ca and P elements decrease.

Figure 9 shows the XRD results to further measure the formed crystalline phase of apatite particles on samples after being soaked in SBF for 3 days. There occur several characteristic diffraction peaks at 26°, 32°, and 39°, corresponding to the (002), (211) and (310) crystalline surfaces of the generated apatite. Additionally, compared to the XRD results before soaking (Figure 2), the diffraction peaks at 21.88° and 23.85° become significantly weak after 3-day soaking, further suggesting the expected generation of apatite covering the PCL substrate. The apatite forming ability of mCNF–BP hybrid monoliths might be highly enhanced by increasing the addition of mCNFs as confirmed in SEM, EDS, and XRD measurement.

### 3.4. In Vitro Biocompatibility

Verifying the biocompatibility of the mCNF–BP hybrids is a prerequisite for in vivo application. The MC3T3-E1 cells were taken as the research object in this experiment. As shown in Figure 10A,B, the cell viability of MC3T3-E1 cells after incubation with 0.5 wt%, 1.0 wt% and 1.5 wt% mCNFs of mCNF–BP hybrid monoliths at various concentration were maintained above 90% for 24 h and 72 h. The results indicated that mCNF–BP hybrids with 0.5 wt%, 1.0 wt% and 1.5 wt% of mCNFs did not induce the toxicity both in 24 h and 72 h for all concentrations. The above results demonstrated that mCNF–BP hybrids were compatible with normal cells.

## 4. Discussion

Based on the various characterizations, applying a simple sol-gel method to mix the mCNFs and BP leads to a unique combination between these two materials at room temperature, which takes as short as 3 days. CNFs are nanomaterials with large aspect ratios, high strength, high modulus, good mechanical properties, and good biocompatibility, making them a promising application in medical biomaterials. However, CNFs have a large number of hydrogen groups on the surface with good hydrophilicity. When combined with some polymers matrix, CNFs and polymers have poor interfacial compatibility due to significant polar differences, and the excellent properties of CNFs cannot be applied. The silane coupling agent KH560 chemically reacts with hydroxides on the CNFs surface and organic polymers, and the chemical junction and charge state on the surface of nanoparticles changed, which can improve the interfacial compatibility and dispersion stability of CNFs with polymers. Furthermore, because of the small particle size and high atom ratio with polymers, improving the bonding effect between the matrix and particles, making it easier to disperse the stress applied, increasing the tensile strength and modulus of the hybrid membranes. The mCNFs have a network due to the entanglement of microfibers, making the PDMS-BG phase stably suspend in the PCL matrix, dramatically reducing preparation time. Additionally, many investigations have found that adding reactive anionic groups to the surface of polymeric materials causes mineral phase crystals to exhibit controlled nucleation and growth characteristics. These findings show that reactive anionic groups (i.e., hydroxyl) on the surface of polymer materials may cause ionic bond adsorption of Ca^2+^ ions and hydrogen bond adsorption of PO_4_^3−^ ions, resulting in local supersaturation and crystal nucleation [27,28]. The hydroxyl groups on the mCNFs materials must thus be able to offer nucleation sites for the formation of mineral crystals, which thereby facilitates the apatite generation. The resulting mCNF–BP exhibits excellent mineralizing and mechanical performance, which can be tailored by incorporating various amounts of mCNFs, and may be an advanced bioactive material in future applications to bone-tissue regeneration.

## 5. Conclusions

The bioactive mCNF–BP hybrids with various amounts of mCNFs incorporation have been produced via a simple sol-gel method. The mCNFs, as the cross-linker and nano-reinforcement filler, can be uniformly dispersed in the BP matrix to accelerate the sol gelation. The mCNF content significantly impacts the structure of the mCNF–BP hybrids, and has proved to be a significant parameter that affects the mechanical properties of the composite hybrids. The resulting mCNF–BP hybrids exhibit improved mechanical performance, including remarkable elastic modulus and toughness. As the mCNFs incorporation increases to 1.0 wt%, the mCNF–BP hybrids exhibit an optimized modulus of elasticity and stress. Additionally, the mCNFs incorporation improves the hydrophilicity of the mCNF–BP hybrids. As a result, the mCNF–BP hybrids soaking in SBF lead to rapid layer-shaped apatite generation on the surface, indicating excellent biomineralization ability while showing good tissue safety. The mCNF–BP hybrids can be expectantly used as implants in medical applications and bone regeneration.

## Figures and Tables

**Figure 1 polymers-15-03497-f001:**
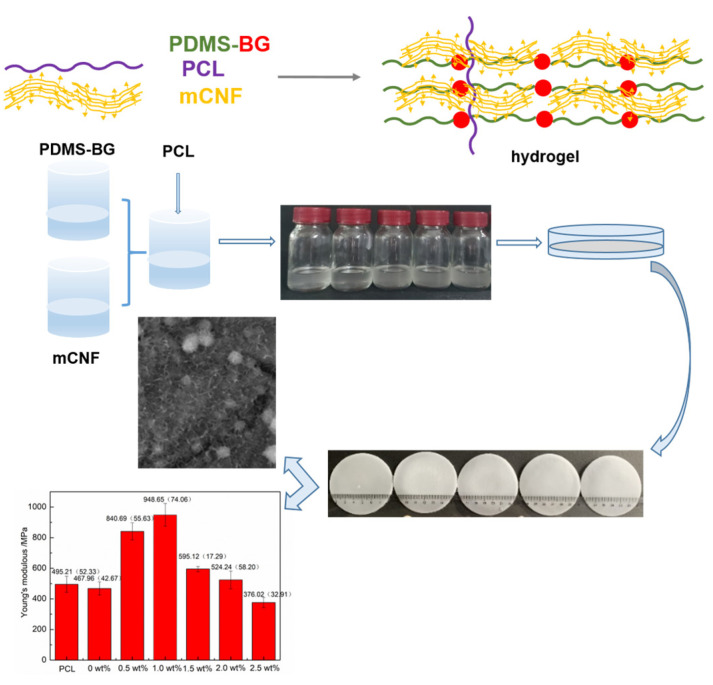
Synthesis process chart of mCNF–BP monolith hybrids.

**Figure 2 polymers-15-03497-f002:**
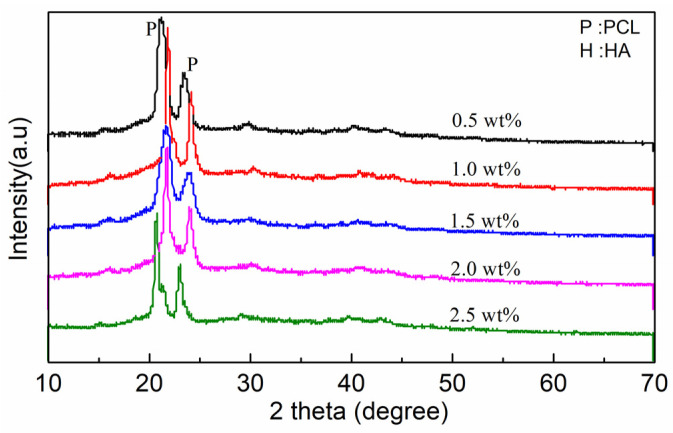
XRD patterns of mCNF–BP hybrids with various amounts of mCNFs incorporation (0.5, 1.0, 1.5, 2.0, and 2.5 wt%). The diffraction peaks of the apatite are marked in the picture.

**Figure 3 polymers-15-03497-f003:**
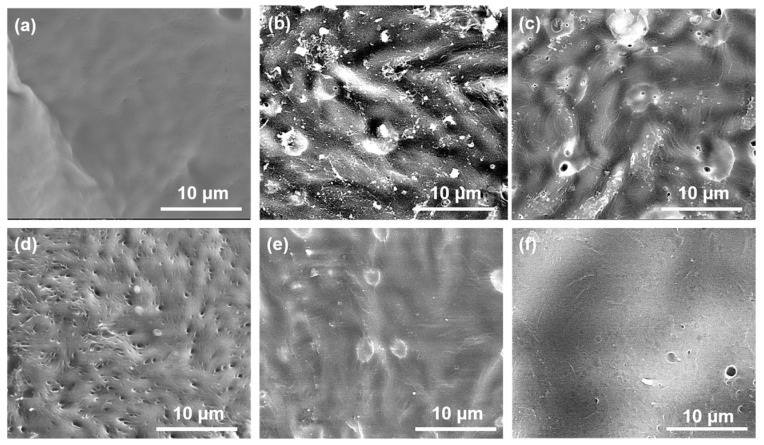
SEM micrographs of mCNF–BP hybrids with various amounts of mCNF incorporation. (**a**) 0 wt%; (**b**) 0.5 wt%; (**c**) 1.0 wt%; (**d**) 1.5 wt%; (**e**) 2.0 wt%; (**f**) 2.5 wt%.

**Figure 4 polymers-15-03497-f004:**
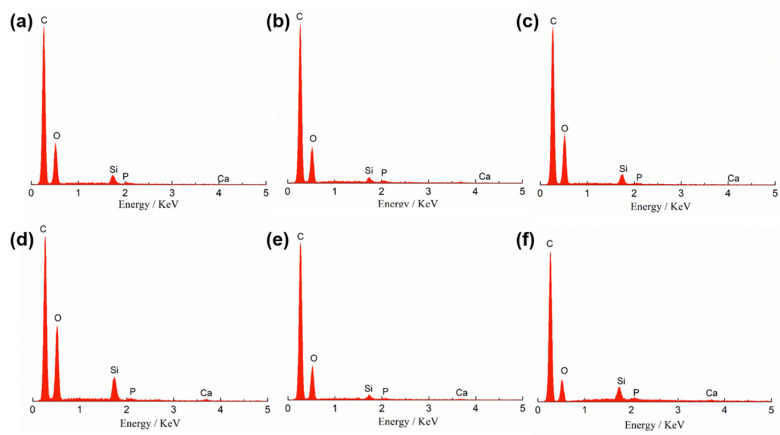
EDS analysis spectra of mCNF–BP hybrids indicate the elemental composition with various amounts of mCNF incorporation. (**a**) 0 wt%; (**b**) 0.5 wt%; (**c**) 1.0 wt%; (**d**) 1.5 wt%; (**e**) 2.0 wt%; (**f**) 2.5 wt%.

**Figure 5 polymers-15-03497-f005:**
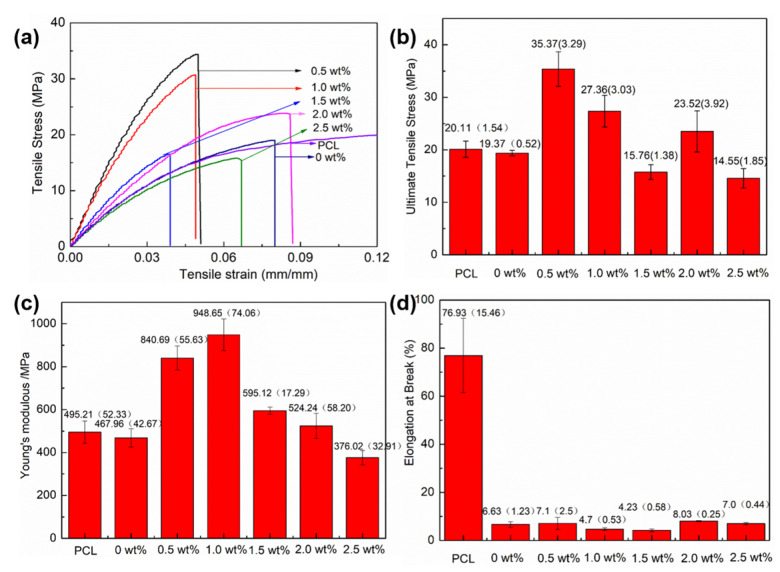
The measurements of the mechanical properties of mCNF–BP with various amounts of mCNFs. (**a**) Stress–strain behavior; (**b**) ultimate tensile strength; (**c**) Young’s modulus; (**d**) elongation at break.

**Figure 6 polymers-15-03497-f006:**
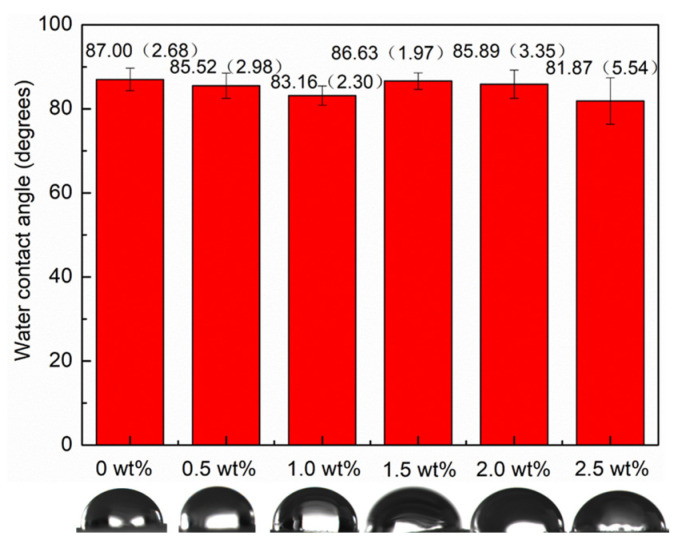
Water contact test of mCNF–BP hybrids with various amounts of mCNFs presenting the hydrophilicity of samples.

**Figure 7 polymers-15-03497-f007:**
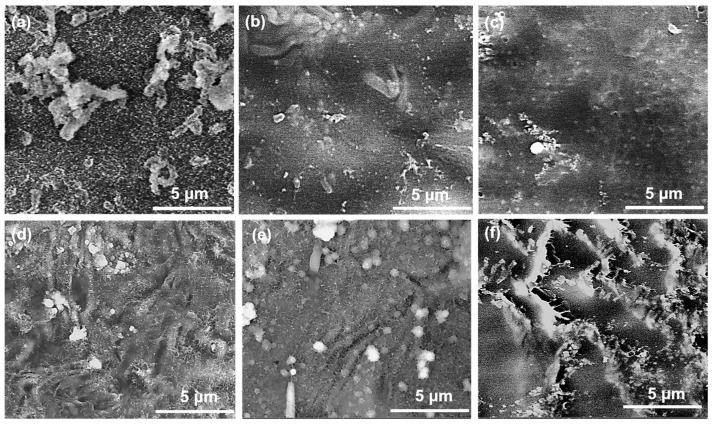
SEM images of the morphology of mCNF–BP hybrid monoliths with various amounts of mCNFs after 3-day soaking in SBF. (**a**) 0 wt%; (**b**) 0.5 wt%; (**c**) 1.0 wt%; (**d**) 1.5 wt%; (**e**) 2.0 wt%; (**f**) 2.5 wt%.

**Figure 8 polymers-15-03497-f008:**
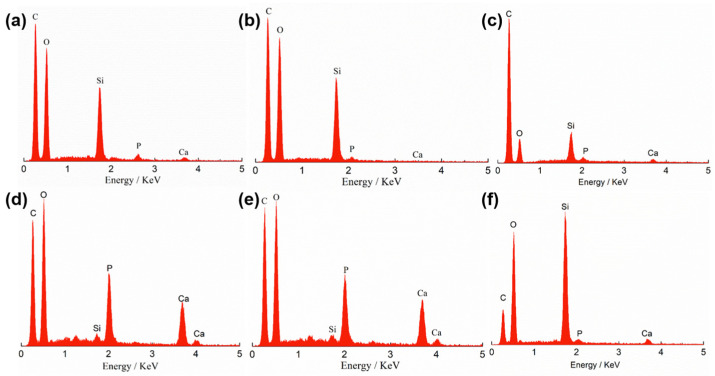
EDS analysis spectra of mCNF–BP hybrid monoliths indicate the elemental composition with various amounts of mCNFs after 3-day soaking in SBF. (**a**) 0 wt%; (**b**) 0.5 wt%; (**c**) 1.0 wt%; (**d**) 1.5 wt%; (**e**) 2.0 wt%; (**f**) 2.5 wt%.

**Figure 9 polymers-15-03497-f009:**
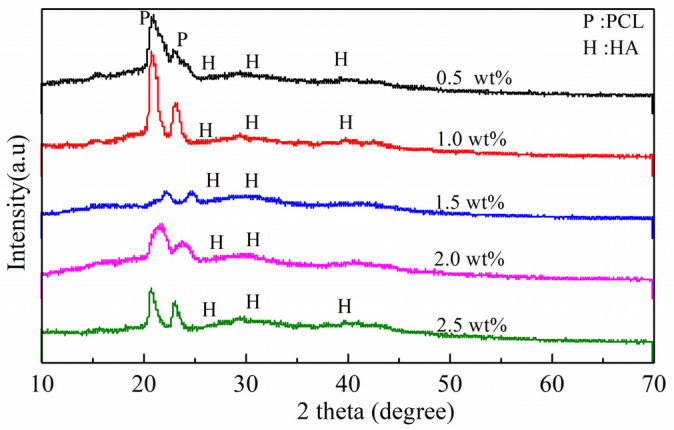
XRD patterns of mCNF–BP hybrids with various amounts of mCNFs (0.5, 1.0, 1.5, 2.0, and 2.5 wt%) after 3-day soaking in SBF. The apatite diffraction peaks were marked in the picture.

**Figure 10 polymers-15-03497-f010:**
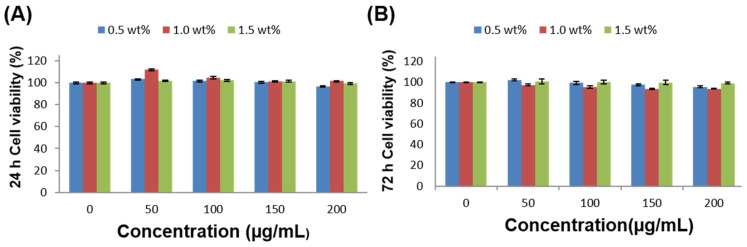
Cellular biocompatibility analysis. (**A**,**B**) Cell viability of MC3T3-E1 cells after incubation with 0.5 wt%, 1.0 wt% and 1.5 wt% mCNFs of mCNF–BP hybrid monoliths at various concentrations for 24 h (n = 4) and 72 h (n = 4).

**Table 1 polymers-15-03497-t001:** Surface chemical composition of the samples before and after soaking in SBF obtained by EDS analysis.

	Si		Ca		P		C		O	
	Mass (%)	Atom (%)	Mass (%)	Atom (%)	Mass (%)	Atom (%)	Mass (%)	Atom (%)	Mass (%)	Atom (%)
0.5 wt%-0d	2.52	1.17	0.19	0.06	0.3	0.13	72	78.25	24.99	20.39
1.0 wt%-0d	2.97	1.39	0.33	0.11	0.54	0.23	70.2	76.91	25.97	21.36
1.5 wt%-0d	5.54	2.7	1.23	0.42	0.45	0.2	60.46	68.99	32.32	27.69
2.0 wt%-0d	1.42	0.66	0.8	0.26	0.43	0.18	73.22	79.29	24.12	19.61
0.5 wt%-3d	3.08	1.43	0.64	0.21	0.55	0.23	73.89	80.31	21.83	17.82
1.0 wt%-3d	1.6	0.75	1.07	0.35	0.76	0.32	71.8	78.3	24.78	20.28
1.5 wt%-3d	1.21	0.75	22.07	9.66	11.33	6.42	31.4	45.88	34	37.29
2.0 wt%-3d	1.11	0.7	21.76	9.56	11.76	6.7	30.48	44.67	34.87	38.37

## Data Availability

All the data supporting the results of the study are included in the paper.

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
