# Peer review of "Strong Bioactive Glass-Based Hybrid Implants with Good Biomineralization Activity Used to Reduce Formation Duration and Improve Biomechanics of Bone Regeneration"

_polymers, 2023, doi:10.3390/polym15173497_

Round 1

Reviewer 1 Report

In the manuscript entitled “Strong Bioactive Glass-based Hybrid implants with Good Bio-mineralization Activity used to reduce formation duration and improve biomechanics of Bone Regeneration” the authors demonstrated the biomineralization efficiency of their developed platforms. The work is interesting and can be considered for publication after addressing the comments given below;

Category: Major Revision

Comments: 1. There is no data to support the interactions within the added components, which influenced the properties of the developed materials. It is recommended to add some characterizations to support the interactions.

2. It is better to mark the formed appetites and m-CNFs in the SEM images. It is also recommended to mentioned the quantitative value of calcium, phosphorus, and other significant contents in the EDX results.   

3. It is advised to add the TEM images to support the mechanical strength of the developed platforms.

4. In the developed samples, which one is suitable for the optimum results according to the authors. Justify it.

5. It is advised to include the cytotoxicity of the developed samples.

6. It is also suggested to perform some more experiments to support the bio-mineralization potential of the developed samples.

Author Response

Responses to reviewers' comments in word document.

Reviewer 2 Report

In this article the authors described the development of bioactive implants based on modified cellulose nanofibers (mCNFs)-reinforced bioactive glass-(BG)polycaprolactone using a modified sol-gel method. The hybrid samples were than characterized in terms of mechanical properties, morphological features, hydrophilicity and biomineralization activity. The manuscript appears to be poor in terms of results and experimental method. The main aim of the work was to improve the mechanical properties of the PCL-BG bioactive materials using mCNFs. However, there are many unclear points and most of all totally absence of interesting results. In particular, the authors did not perform any characterization from biological point of view, so the results obtained are not applicable in the biomaterials field.  I would suggest the rejection of the manuscript in polymers journal and the reconsideration only after a major revision. In report here my comment about the manuscript:

1.      In the introduction part it is not clear which is the advancement compared to the previous work of the authors.

2.      On one side the authors claim to add the mCNFs to improve the mechanical properties of the PCL on the other side to improve the hydrophilicity. However, to improve the interface between the PCL and the mCNFs a silane coupling agent have been adopted. However, the coupling agent make the mCNFs less hydrophilic. So, in my opinion it is a nonsense using a coupling agent to improve the interface and decrease the hydrophilicity.

3.      From the mechanical characterization at pp 6 line 191 the authors said, “the samples exhibit a uniform elastomeric behaviour”. However, from the stress-strain curve the behaviour is not elastomeric at all but elastic-brittle.

4.      Moreover, from the results of the mechanical properties if on one side the young modulus increased by using 1.0 wt% of mCNFs the elongation at break strongly decreased making the material highly brittle. Is this an advantage?

5.      How did the authors explain the decrease in the mechanical properties by using an amount of mCNFs higher than 1.0wt%.

6.      Figure 3 has really poor quality please use SEM with a better resolution and focus.

7.      The hydrophilicity test reported at pp 7 clearly showed that the presence of mCNFs did not modify the water contact angle at all. So, what is the meaning of using the mCNFs?

8.      In the discussion part the authors clearly claims that the designed materials have promising application in medical biomaterials and can found application in bone engineering. How can they claim this without any biological characterization as: cytotoxicity, cells-material interaction, cells proliferation?

The English need a thorough editing by a mother tongue due to many sentence written with a poor English language.

Author Response

(The authors gave the same response as above.)

Reviewer 3 Report

The subject of the research it is an interesting one. There are some remarks for the authors:

1. line 45: please do not begin with AND the sentence.

2. line 51: the syllabification is not correct in case of the word "developed" 

3. chapter 3.2 Mechanical tests: the authors should compare the measured values of tensile stress and Young's modulus of other biocomposites!

4. Fig.9. The XRD spectra of the samples show amorf substances, they could not be used to prove the existence of apatite phase! This is the main weakness of the paper! The authors has to be more careful by proving the formation of new apatite layers and they need other analyses to demonstrate the formation of apatite in SBF on the samples surfaces.

Author Response

(The authors gave the same response as above.)

Round 2

Reviewer 1 Report

Accepted.

Author Response

Thank you for the comments received on the article. Thank you for reviewing my article and providing particularly valuable comments to make our work better.

Reviewer 2 Report

The authors did not reply in detail to the reviewer comments and the manuscript still has some scientific flaws. In my opinion the manuscript is still not suitable for publication because the proposed approach has very low scientific soundness and absence of interesting results.

Reply to comment 1: ok.

Reply to comment 2: the reply of the authors is not so clear. From Figure 6 the water contact test of mCNFs-BP samples containing 0wt% of mCNFs is 87°. The addition of mCNFs did not change drastically the hydrophilicity of the materials and moreover is did not show a linear behavior. In fact, the water contact angle decreases with 1,0wt% and increase with 1,5 wt%. This means that the presence of mCNFs doe no alter the hydrophilicity of the material.

Reply to comment 3: ok.

Reply to comment 4: the authors did not reply to the reviewer comment. In my opinion add a filler that drastically decrease the elongation at break of the material is not an improvement even if the elastic modulus raises. This is a serious flaw of then proposed approach.

Reply to comment 5: the explanation is reasonable, but some morphological analysis is needed to support the authors thesis about the agglomeration of the mCNFs.

Reply to comment 6: Usually to perform SEM analysis of polymers the sample is coated with a thin layer of gold-palladium to improve the conductivity and avoid damage on the sample surface. Please provide image with better quality and more explicative because from Figure 3 it is not possible to detect any differences in the sample surface increasing the amount of mCNFs. Moreover, the authors should use the SEM analysis to analyze the distribution and morphology of the mCNF.s fillers inside the PCL matrix.

Reply to comment 7: the same reported in the reply to comment 2.

Reply to comment 8: regarding the in vitro cytotoxicity test it is not clear how the experiment has been conducted. In particular it is not clear in Figure 8 what represent “concentration” on the x-axis? Moreover, where is the “control” in the graph

Author Response

Already answered in the document.

Reviewer 3 Report

Thank you for your answers and corrections. Please introduce in the paper text also the answer regarding the formation of new apatite layer (poor crystallinity of HAP and stoichiometry proved by EDS).

Author Response

Already answered in the document.
